# Goal-Setting in Multiple Sclerosis-Related Spasticity Treated with Botulinum Toxin: The GASEPTOX Study

**DOI:** 10.3390/toxins14090582

**Published:** 2022-08-24

**Authors:** Ines Baccouche, Djamel Bensmail, Emilie Leblong, Bastien Fraudet, Claire Aymard, Victorine Quintaine, Sandra Pottier, Thibaud Lansaman, Claire Malot, Philippe Gallien, Jonathan Levy

**Affiliations:** 1Department of Physical Medicine and Rehabilitation, Raymond-Poincaré University Hospital, AP-HP, 104 Blvd Raymond Poincaré, 92380 Garches, France; 2Fondation Garches, 104 Blvd Raymond Poincaré, 92380 Garches, France; 3Pôle MPR St. Hélier—Living Lab ISAR, 54 Rue St. Hélier, CEDEX, 35043 Rennes, France; 4Fondation Sainte Marie Paris, 167 Rue Raymond Losserand, 75014 Paris, France

**Keywords:** spasticity, botulinum toxin, multiple sclerosis, disability, locomotion, goal attainment scale

## Abstract

Spasticity is one of the most disabling symptoms in multiple sclerosis (MS). Botulinum toxin injection (BTI) is a first-line treatment for focal spasticity. There is a lack of evidence of a functional improvement following BTI in MS-related spasticity. To describe goal-setting for BTI in MS, and evaluate the degree of attainment, using goal attainment scaling (GAS) 4-to-6 weeks after injection session, a one-year multi-center retrospective observational study assessing goal-setting and achievement during BTI session in spastic patients with MS was set up. Following the GAS method, patients and their physicians set up to three goals and scored their achievement 4 to 6 weeks thereafter. Commonly used goals from three centers were combined into a standardized list and 125 single BTI sessions were analyzed. The most frequent goals regarded lower limb (LL) impairments (equinovarus foot, toe claw) or locomotion (stability, walking distance, clinging) and accounted for 89.1%, versus 10.9% for upper limb (UL), mostly for mild-to-moderate MS. Overall, goals were frequently achieved (85.77%) mainly when related to gait and mobility rather than hygiene and ease of care. This study gives an overview on the most frequent, relevant, and achievable goals to be set in real-life practice of BTI for spasticity management in MS.

## 1. Introduction

Multiple sclerosis (MS) is an autoimmune disease that affects more than 3.2 million people worldwide and is considered the most frequent cause of acquired disability among young adults. Inflammation of the neuraxis causes demyelination with what is called spatial and temporal dissemination. Various symptoms can occur along the course of the disease such as fatigue, bladder/bowel disorder, visual problem, sexual dysfunction, cognitive and emotional disorder, tremor, ataxia, pain, and spasticity leading to progressive neurologic and functional disabilities and a decline in patient’s quality of life [1,2,3].

Spasticity is one of the most disabling symptoms in MS. It has been defined by Lance in 1980 as “a motor disorder characterized by a velocity-dependent increase in the tonic stretch reflex (muscle tone) with exaggerated tendon jerks, resulting from hyperexcitability of the stretch reflex, as one component of the upper motor neuron syndrome” [4]. A new definition has been introduced more recently by Pandyan and is now considered the most appropriate. It encompasses more clinical aspects of spasticity than the previous: “Spasticity is disordered sensorimotor control, resulting from an upper motor neuron lesion, presenting as intermittent or sustained involuntary activation of muscles” [5]. It is associated with pain, limited range of motion or abnormal postures, and its severity follows disease evolution [6,7]. The most common consequences of spasticity are difficulties in limb mobilization, pain, grasping impairment, positioning problems either in bed or wheelchair and gait disturbance. 

The most frequent discomforts in the lower limbs when walking are toe clawing during stance phase, equinus or varus-equinus causing the foot to scratch on the ground, knee recurvatum and thigh adductum, hence disrupted balance [8]. In the upper limbs, spasticity mainly affects the flexors of the fingers and elbow preventing the opening of the hand, the extension of the limb and its separation from the body making it difficult to grip, dress, wash, and transfer [9].

Type A botulinum neurotoxin (BoNT-A) is the most effective treatment for focal spasticity and has been approved by most national medication agencies worldwide in MS since 1989 [10]. A small number of studies explored specifically the use of BT in MS-related spasticity, mostly for lower limb [11,12,13,14] and only two included upper limb [15,16]. All have shown evidence of BoNT-A efficiency for focal spasticity with a decrease of at least 1 point on the Modified Ashworth Scale (MAS).

However, there is a lack of properly demonstrated evidence of a functional improvement following botulinum toxin injection (BTI) [17]. In fact, mild improvements in impairment-related goals translated barely into functional gain, except for passive functional goals such as nursing [17,18]. This has only been shown in post-stroke spasticity.

The first step in demonstrating functional efficacy of BTI is to identify clearly what the objectives of treatment are for patients and physicians in real-life practice. This has been extensively studied by Turner-Stokes and Ashford for post-stroke spasticity. They categorized patients’ goals into the ICF framework, i.e., symptoms/impairments and activities/participation which included passive (i.e., related to the ease of care by a third) and active functional goals [19,20,21,22,23,24,25]. Most frequent goals were related to impairment and passive function [18,19], whereas active function was cited for a subpopulation of patients with minor impairment, and after multiple BTI cycles [20,22]. All this underlines the importance of pre-identifying realistic accurate goals according to patients’ status. 

Alongside with standardized outcome assessment, the Goal Attainment Scaling (GAS) is a technique used to set realistic individualized goals with specific qualitative information by patients and health professionals and to quantify their achievement after a predetermined period [23,24]. First introduced by Kiresuk and Sherman [25] in mental health and then completed with Turner-Stokes guide [26] for rehabilitation use [24], this method provides an objective evaluation based on subjective goals and has shown to provide a higher sensitivity to functional gains after treatment than standard measures [27,28,29].

Different from stroke, MS is a progressive disease with multiple functional consequences of related spasticity, and affecting mostly young adults [9]. We first hypothesized that objectives for the treatment of focal spasticity with intramuscular BTI, may differ for patients with MS compared to post-stroke patients, and needed to be listed independently from the latter. Second, we inferred that knowing these objectives could help and ease GAS scoring in clinical and research practice in MS-related spasticity, in order to assess the efficacy of BTI in this indication, and therefore help physicians to set realistic goals with their patients. The main purpose of this study was therefore to identify the therapeutic goals set by patients and their physicians for the treatment of focal spasticity with BTI in MS. Second, we aimed to evaluate the achievement of these goals 4 to 6 weeks after an injection session.

## 2. Results

### 2.1. Patient Characteristics

One hundred and twenty-five single-patient BTI sessions were assessed for final analysis by the end of a one-year period of monitoring (from July 2020 to June 2021). The main patients’ characteristics are reported in Table 1. Of note, no difference for the main descriptive data between centers, such as disease evolution, severity, and course of MS existed. Of all patients, 45 had oral antispastic treatment (all had baclofen and only two were associated with dantrolen) and 80 did not have antispastic treatment. No significant difference was noted (*p* > 0.05) which confirms a homogeneous study population.

### 2.2. Description of Goal-Setting

For all 125 patients there was at least one goal set prior to BTI. Almost three quarters (72.8%) defined a second goal, and only 48% set three goals prior to BTI.

Overall goal setting showed that most aimed either to reduce the clinging of the ground, or to improve stability and static balance during ground support (both 9.87%), followed by the will to reduce toe claw to facilitate the gait for 7.73%, and finally 6.67% to reduce an equinus or equinovarus foot disturbing gait (Table 2).

Taken solely, the primary GAS goal (goal 1) was mostly to reduce the clinging of the ground (18.4%), followed by improving stability in standing position and static balance (16.8% + 2.4%), and third to improve walking distance, or to reduce an equinus disturbing gait (both 8%). Exhaustive description of primary, secondary, and tertiary set goals is presented in Table 3. 

### 2.3. Goals Distribution by Lower or Upper Limb

The overall analysis showed that most set goals regarded lower limb symptoms or function (89.1% vs. 10.9% for upper limbs). Similar results were observed in both centers taken separately. For lower EDSS scores (i.e., the least severe forms of MS) set goals concerned exclusively LL. For moderate disability (EDSS 6 to 7.5) goals concerned mostly the LL 89.8% versus 10.2% for UL. Finally for the most disabled patients, goals were mostly set to improve upper limb symptoms or function (55.9 vs. 44.1%). Full description of LL/UL repartition is detailed in Table 4.

In general, among UL goals the most frequent were to prevent or limit muscle contractures, the second were to reduce pain during mobilization and equally, to ease shoulder abduction. Most frequent LL goals were the ones previously described.

### 2.4. Goals Description According to Descriptive Data

Frequent selected goals were roughly the same in both sexes with a slight difference in the order of frequency (Table 5).

Physicians in center 1 tended to aim at first the improvement of stability, second the reduction of toe claw, and third the clinging to the ground. In center 2, the improvement of walking perimeter tended to be preferred to reduction of the clinging and last balance improvement (Table 6).

### 2.5. BoNT-A Efficacy and Goal Attainment

A total of 264 goals were set, and 239 were properly assessed at 4 to 6 weeks for achievement or not. Of all, 205 were achieved (85.77%) and only 34 were not (14.23%).

For upper limb selected goals and for lower limbs, 88% (22/25) and 86% (183/214) were achieved respectively. Except unset goals, all others were at least 50% attained. 16 goals were 100% achieved including 9 lower limb goals and 7 upper limb goals. The full description of goals by decreasing order starting from the most frequently achieved goal, is provided in Table 6. 

### 2.6. Pre-Post Spasticity Assessment (MAS)

The T test shows that the mean of the MAS score between PRE and POST injection were significantly different (2.77 in PRE vs. 1.97 in POST; *p* < 0.05—Figure 1), with an overall decrease of muscle tone by one point.

## 3. Discussion

This study gives an overview on the most relevant, frequent, and achievable goals to be set in real-life practice of BTI for spasticity in MS.

Our findings show that objectives of BTI for MS patients differ from post-stroke patients [22,30,31]. In this population, studies have mainly explored UL-related goals, and patients suffering from post-stroke spasticity tended to prefer passive function goals and pain objectives, rather than improving active function. In our study focusing on MS, most set goals were related to lower limbs spasticity, and furthermore to spastic schemes of distal muscles (i.e., equinus and equinovarus foot, and/or toe-claws). Goals related to active function (i.e., walking) prevailed on passive function. This remained true for mild to moderate forms of MS (EDSS scores ≤ 7.5) We hereby demonstrated that lower limb function may be the main concern for patients with MS and their physicians, until complete loss of locomotor abilities, even if upper limb function could already be impaired [32,33]. Hygiene related goals (i.e., passive function) were rarely selected and concerned mostly reduction of a hip adductum hampering intimate hygiene and dressing. Shoeing facilitation, facilitation of dressing, toilet, nursing, and reduction of hand maceration for upper limb spasticity, are major concerns for the most disabled patients suffering from MS [8,9,11,13]. Hygiene objectives in both LL and UL may be achieved differently, and as spasticity increases over time in these patients, one may consider injecting BoNT-A in UL and treat LL and trunk spasticity with treatments such as intrathecal baclofen therapy [34].

We intended to use the GAS scoring as secondary outcome in order to verify the efficacy of BTI on a functional point of view. First, we confirmed in our cohort, a general significant spasticity reduction by comparing MAS score before and 4 to 6 weeks after BTI. Hence, the study meets the standards required to discuss GAS results. We chose the GAS method for its sensitivity in assessing subjective effects of similar therapeutics used in a homogenous population, in real life practice, and as it involves both patients and clinicians in the process [28]. The purpose of our study being observational and descriptive, we chose to verify which goals were more frequently achieved without taking into account the level of achievement, and dichotomized goal achievement as ‘achieved or not’ (i.e., GAS light methods).

In this respect, most of our study goals were achieved whether they were related to upper limbs or lower limbs with a high frequency of achievement in UL symptom-related goals and LL symptoms and function-related goals. For example, for UL goals, the reduction of muscle-tendon retraction and the improvement of shoulder abduction or hand opening were achieved every time they were set prior to BTI. For LL, the reduction of equinovarus or *extensor hallucis longus* dystonia, were achieved in 100% of cases. Interestingly, toe claw or clinging were less likely to be achieved (88 and 79% of cases respectively), whereas overall walking was achieved in 100% of cases. This confirms that, apart from the reduction of symptoms [35], the treatment tends to meet the functional expected improvement. Improved active functions mainly covered gait and walking parameters that could easily be quantifiable for the least severe patients [36] followed by less frequent passive hygiene, positioning, and dressing functions for severe non ambulatory patients. Frequently achieved functional goals from our pre-established list are relevant functional goals for future interventional studies as baseline references, to verify BTI efficacy.

A major limitation for this second part of the study would be the use of a pre-established list of goals that limits possible choices and may not lead to an exhaustive list. A goal setting protocol, described by Turner-Stokes [19] allowed a free description of goals by the patient structured by the clinician according to specific criteria. In our study, a previous list with an open-field for goal setting was proposed, but at final data collection and analysis, all reported goals fitted within the consensual list. However, the overall purpose of the GASEPTOX project is to provide caregivers and physicians involved in the treatment of spasticity, with realistic goals to establish with patients prior to BTI. Goals should be set in accordance with the SMART approach, hence being specific, measurable, achievable, relevant, and timed [37]. Even if we kept in mind this recommendation while writing our protocol and article, we admit that the design of our study could not allow a properly SMART approach, especially for measurement and objective assessment. A nationwide survey assessing practice of goal setting by physicians should complete the main outcome of the present study. Answers could be compared to the goal list that has been proposed by referral expert in the field in this study. Future interventional studies assessing the functional efficacy of BTI to treat MS-related spasticity, should take these results into account. Some goals could be clustered, and the total number of goals reduced to allow a proper SMART approach.

Another limitation is the absence of sample size calculation that prevented any generalization of our findings for the second outcome. Furthermore, as this study was not designed to assess either overall or functional efficacy of BoNTA, we could not follow a proper GAS method, as well as a shorter version of it [24,29]. We followed common habits of our three centers. This means that the assessment of goal achievement was based on patients’ declarations rather than objective and quantitative assessment. If no conclusion could be drawn from our study regarding BTI effectiveness to treat MS-related spasticity, we believe we set strong bases for future prospective interventional trials. Such trials should respect guidelines and resemble previously published protocols for stroke-related spasticity [19,37,38,39,40]. However, the high level of subjective attainment in our study could be explained by a 20 year long experience of the physicians involved in this study. We may infer that appropriate goal-setting and achievement could be experience-related and hereby not represent real-life practice. We do think this study could help less-experienced doctors set appropriate and achievable goals for BTI in MS related spasticity.

## 4. Conclusions

In conclusion, goal setting should be necessary prior to the start of BTI for the treatment of spasticity in MS. Indeed, knowing which goals are pertinent and realistic as they are most likely to be achieved is mandatory to gain therapeutic alliance. In mild to moderate MS, targeting muscle groups responsible for equinovarus foot and toe clawing may help to improve lower limb function, especially patient perception of gait quality. In most severe patients, hygiene is a priority target but less frequently achieved, which may invite the physician to consider therapeutic alternatives in these patients.

## 5. Future Perspectives

This study is a preliminary study of the GASEPTOX project. Future studies should be clinical trials assessing as primary outcome, the functional efficacy of BTI in patients suffering from MS-related spasticity. The use of *a priori* standardized list of pertinent goals, a proper GAS method, and precise assessment and measurement of each goal, are mandatory in order to demonstrate the efficacy of BTI on function goals in this population.

## 6. Methods

### 6.1. Study Design and Setting

This was a retrospective observational study, based on real-life practice for a one-year duration, in the spasticity clinic of three French tertiary hospitals. 

### 6.2. Study Population

Data from consecutive BTI sessions regarding all adult patients with a confirmed diagnosis of MS [41] and followed or addressed in one of these three spasticity clinics were analyzed in this study. Exclusion criteria were the inability of the patient to understand and establish functional goals with their physician, rejection or non-indication of the treatment with BoNT-A, and *a posteriori* objection to the use of personal medical information.

### 6.3. Defining Therapeutic Goals

Physicians from all three physical and rehabilitation medicine departments commonly use goal definition and assessment for the follow-up of treatment with BTI. Four of them (DB, PG, CA, JL) considered as experts in the field of spasticity, were asked to provide the most frequent goals retrieved from patients’ medical charts. Then, they consensually defined a list that encompassed pertinent goals to be used in this analysis. The investigators referred, if possible to the SMART approach to retain goals that are specific, measurable, achievable, relevant, and timed [37]. Goals were listed according to physicians’ experience and previously published data on spasticity caused by other etiologies. Twelve concerned the upper limbs, symptoms or function, and 24 the lower limbs. The usual habit in all three centers is to establish with the patient a list of up to three objectives prior to BTI. The achievement of these goals is regularly assessed at the follow-up consultation.

### 6.4. Botulinum Toxin Injections and Follow-Up

There was no specific intervention in the context of the study and BTI were performed as usual by each physician according to the patients’ needs with either one of the three BoNT-A formulations available in France (i.e., abobotulinum, incobotulinum, and onabotulinum toxins). Patients pursued their usual physiotherapy and rehabilitation alongside as they were accustomed to (mostly community-based or in regular day-hospital setting). Patients were regularly seen at follow-up consult 4 to 6 weeks after injection; all of this was in accordance with the national guidelines from the French Society of Physical and Rehabilitation Medicine [42].

### 6.5. Data Collection

Main data consisted of goals set by the physicians and their patients. Clinical data were also collected when available such as: gender, date of birth, date of MS diagnosis, expanded disability status scale (EDSS score) [43] assessing the importance of impairment and functional loss related to MS defining disease stages, the course of MS, spasticity assessment using the modified Ashworth scale (MAS) [44], and the type and doses of BoNT- A. The assessment of goal achievement was collected according to the GAS-light method, which dichotomize each goal as achieved vs. not achieved [29].

### 6.6. Statistical Analysis

Descriptive data were expressed as means ± standard deviation when applicable, and as medians with interquartile ranges when they did not pass normality tests, or for discontinuous variables. Pre/post spasticity assessed with the MAS was expressed as means (after standardization into a 5-point scale) and compared using Student *t*-test. Goal achievement was expressed as percentage of goals achieved in the overall population and diverse subgroups.

### 6.7. Ethics

This study with the related data collection, patient information, and analyses were performed in accordance with the French bioethics’ laws regarding medical research. It followed a reference methodology for non-interventional research (MR004) registered and approved under the number 2213172v0 by the Commission Nationale Informatique et Libertés on 2 May 2019.

## Figures and Tables

**Figure 1 toxins-14-00582-f001:**
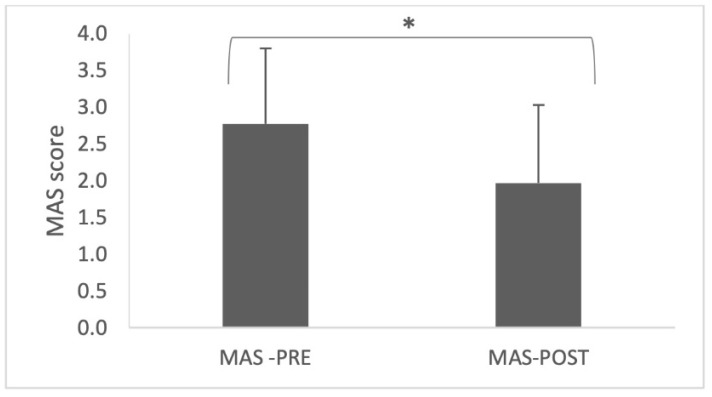
Pre/Post botulinum toxin injection spasticity assessment with mean modified Ashworth scale (MAS). (*) indicates a statistically significant difference (*p* < 0.05).

**Table 1 toxins-14-00582-t001:** Means, medians, standard deviations of patient characteristics in center 1 and center 2.

		General	Center 1	Center 2
N		125	81	44
Sex		36 male 85 female	25 male 52 female	11 male 33 female
Age (year)	Mean ± SD	54.9 ± 11.1	56.0 ± 10.4	53.0 ± 12.0
Disease evolution (year)	Mean ± SD	18.8 ± 9.7	18.8 ± 9.6	19 ±10.1
EDSS	median	6	6	6
	IQ	[4; 6.5]	[4.5; 6.5]	[3; 7]
MS phenotype		SP: 73 PP: 22 RR: 30	SP: 51 PP: 15 RR: 15	SP: 22 PP: 7 RR: 15

Legend: IQ = interquartile range; MS = multiple sclerosis; PP = primary progressive; RR = relapsing remitting; SD = standard deviation; SP = secondary progressive.

**Table 2 toxins-14-00582-t002:** Occurrence frequency of selected goals when primary and secondary GAS goals combined.

Goals	Percentage
Reduce the clinging of the ground	9.87%
Improve stability during ground support	9.87%
Reduce a toe claw to ease gait	7.73%
Reduce an equinus disturbing gait	6.67%
Improve walking perimeter	4.00%
Reduce a knee recurvatum in the support phase	3.47%
Other LL	3.20%
Improve balance	2.93%
Reduce a disturbing dystonia in hallux extension	2.40%
Reduce an equinus varus disturbing gait	2.40%
Prevent or limit retractions	1.87%
Improve walking speed	1.87%
Facilitate transfers	1.87%
Reduce spontaneous pain or mobilization pain	1.60%
Reduce knee flessum	1.60%
Reduce painful toe claw	1.33%
Facilitate the positioning of the feet on the wheelchair pallets	1.07%
Reduce a hip adductum disturbing the toilet, the dressing…	1.07%
Prevent or limit retractions	1.07%
Reduce spontaneous or mobilization pain	1.07%
Facilitate shoulder abduction	0.80%
Facilitate the use of the hand as an auxiliary hand	0.80%
Improve knee flexion in the oscillating phase	0.80%
Other UL	0.80%
Facilitate the passive opening of the hand	0.53%
Facilitate dressing, hygiene, nursing	0.53%
Reduce a disturbing attitude of the upper limb when walking or in other situations	0.53%
Reduce a hip adductum disturbing gait	0.53%
Facilitate the active opening of the hand to improve prehension	0.27%
Reduce hand maceration	0.27%
Reduce access to walking aids	0.27%
Facilitate knee flexion in various situations such as sitting in the wheelchair	0.27%
Facilitate shoeing	0.27%
Facilitate the use of the hand in other activities (computer keyboard, tablets…)	0.00%
Facilitate the wearing of a resting orthosis	0.00%
Facilitate access to the perineum (probes, sexuality, toilet, hygiene…)	0.00%

Legend: LL = lower limb; UL = upper limb.

**Table 3 toxins-14-00582-t003:** Occurrence frequency of selected primary, secondary, and tertiary GAS goals.

	GAS Goal 1 Percentage	GAS Goal 2 Percentage	GAS Goal 3 Percentage
Facilitate shoulder abduction	1.6%	0.8%	
Facilitate the active opening of the hand to improve prehension	0.8%		
Facilitate the use of the hand in other activities (computer keyboard, tablets…)			
Facilitating the use of the hand as an auxiliary hand	0.8%	1.6%	
Facilitate the passive opening of the hand	1.6%		
Reduce hand maceration		0.8%	
Prevent or limit retractions	2.4%	2.4%	0.8%
Facilitate the wearing of a resting orthosis			
Facilitate dressing, hygiene, nursing	0.8%		0.8%
Reduce spontaneous or mobilization pain	1.6%	1.6%	1.6%
Reduce a disturbing attitude of the upper limb when walking or in other situations	0.8%		0.8%
Reduce the clinging of the ground	18.4%	8.8%	2.4%
To improve balance	2.4%	4.8%	1.6%
To improve stability during ground support	16.8%	10.4%	2.4%
Improve walking speed	2.4%	1.6%	1.6%
Improve walking perimeter	8.0%	2.4%	1.6%
Reduce access to walking aids		0.8%	
Reduce a toe claw to ease gait	5.6%	4.0%	13.6%
Reduce painful toe claw	1.6%	0.8%	1.6%
Reduce a disturbing dystonia in extension of the hallux	3.2%	1.6%	2.4%
Reduce an equinus disturbing gait	8.0%	9.6%	2.4%
Reduce an equinus varus disturbing gait	2.4%	3.2%	1.6%
Facilitate the positioning of the feet on the wheelchair pallets		1.6%	1.6%
Reduce a hip adductum disturbing gait	1.6%		
Reduce a hip adductum disturbing the toilet, dressing…	2.4%	0.8%	
Facilitate access to the perineum (probes, sexuality, toilet, hygiene…)			
Reduce a knee recurvatum in the support phase	2.4%	6.4%	1.6%
Improve knee flexion in the oscillating phase	0.8%	1.6%	
Facilitate knee flexion in various situations such as sitting in the wheelchair	0.8%		
Reduce knee flessum	4.0%	0.8%	
Facilitate shoeing		0.8%	
Facilitate transfers	0.8%	2.4%	2.4%
Prevent or limit retractions	0.8%	1.6%	0.8%
Reduce spontaneous or mobilization pain	1.6%	0.8%	0.8%
Other UL	2.4%		
Other LL	3.2%	0.8%	5.6%

Legend: GAS = goal attainment scaling; LL = lower limb; UL = upper limb. Goals 1, 2 and 3 related to the number of goals set by patients in decreasing priority (1 being most important).

**Table 4 toxins-14-00582-t004:** Distribution of lower and upper limb selected goals by center, EDSS score, sex, and form of MS.

		LL	UL
General		89.1%	10.9%
Center	Center 1 Center 2	88.8% 90.2%	11.2% 9.8%
EDSS	[2; 5.5] [6; 7.5] [8; 8.5]	100% 89.8% 44.1%	0% 10.2% 55.9%
Sex	M F	88.8% 88.6%	11.2% 11.4%
MS phenotype	PP RR SP	85.1% 100% 86.3%	14.9% 0% 13.7%

Legend: EDSS = expanded disability status scale; LL = lower limb; MS = multiple sclerosis; PP = primary progressive; RR = relapsing remitting; SP = secondary progressive; UL = upper limb.

**Table 5 toxins-14-00582-t005:** Percentages of most frequent goals by sex, center, EDSS, and phenotype of MS.

Sex	Male			Female			
	Reduce a toe claw to ease gait		12.0%	Improve stability during ground support			9.8%
	Reduce the clinging of the ground		10.2%	Reduce the clinging of the ground			9.0%
	Improve stability during ground support		9.3%	Reduce an equinus disturbing gait			6.7%
Center	**Center 1**			**Center 2**			
	Improve stability during ground support		14.8%	Improve walking perimeter			9.1%
	Reduce a toe claw to ease the gait		11.9%	Reduce the clinging of the ground			6.8%
	Reduce the clinging of the ground		11.5%	Improve balance			3.8%
EDSS	**[2; 5.5]**		**[6; 7.5]**			**[8; 8.5]**	
	Reduce the clinging of the ground	16.1%	Improve stability when ground support		11.3%	Prevent or limit retractions	12.5%
	Improve stability during ground support	11.5%	Reduce a toe claw to ease gait		9.3%	Other UL	8.3%
	Reduce a toe claw to ease the gait	8.6%	Reduce an equinus disturbing gait		8.7%	Facilitate shoulder abduction	6.3%
**MS Phenotype**	**PP**		**RR**			**SP**	
	Reduce the clinging of the ground	16.4%	Reduce the clinging of the ground		16.7%	Improve stability during ground support	11.4%
	Improve stability during ground support	9.1%	Improve stability during ground support		6.7%	Reduce a toe claw to ease the gait	8.2%
	Reduce a toe claw to ease the gait	9.1%	Improve walking perimeter		6.7%	Reduce the clinging of the ground	6.8%

Legend: EDSS = expanded disability status scale; LL = lower limb; MS = multiple sclerosis; PP = primary progressive; RR = relapsing remitting; SP = secondary progressive; UL = upper limb.

**Table 6 toxins-14-00582-t006:** Goal achievement frequency.

Goals	UL/LL	Number of Goal Selected Times	Number of Goal Achieved Times	Percentage
Improve walking perimeter	LL	12	12	100%
Reduce an equinus varus disturbing gait	LL	9	9	100%
Reduce a disturbing dystonia in hallux extension	LL	8	8	100%
Reduce knee flessum	LL	6	6	100%
Prevent or limit retractions	UL	5	5	100%
Prevent or limit retractions	LL	4	4	100%
Facilitate shoulder abduction	UL	3	3	100%
other	UL	3	3	100%
Facilitate dressing, hygiene, nursing	UL	2	2	100%
Reduce a hip adductum disturbing gait	LL	2	2	100%
Facilitate the active opening of the hand to improve prehension	UL	1	1	100%
Facilitate the passive opening of the hand	UL	1	1	100%
Reduce hand maceration	UL	1	1	100%
Reduce access to walking aids	LL	1	1	100%
Facilitate knee flexion in various situations such as sitting in the wheelchair	LL	1	1	100%
Facilitate shoeing	LL	1	1	100%
Improve stability during ground support	LL	31	29	93.5%
other	LL	11	10	90.9%
Reduce a toe claw to ease the gait	LL	26	23	88.5%
Facilitate transfers	LL	7	6	85.7%
Reduce an equinus disturbing gait	LL	23	19	82.6%
Reduce painful toe claw	LL	5	4	80%
Reduce the clinging of the ground	LL	29	23	79.3%
Reduce spontaneous pain or mobilization pain	UL	4	3	75%
Facilitate the positioning of the feet on the wheelchair pallets	LL	4	3	75%
Reduce a hip adductum disturbing the toilet, the dressing…	LL	4	3	75%
Improve walking speed	LL	6	4	66.7%
Facilitate the use of the hand as an auxiliary hand	UL	3	2	66.7%
Reduce spontaneous or mobilization pain	LL	3	2	66.7%
Reduce a knee recurvatum in the support phase	LL	11	7	63.6%
Improve balance	LL	8	5	62.5%
Reduce a disturbing attitude of the upper limb when walking or in other situations	UL	2	1	50%
Improve knee flexion in the oscillating phase	LL	2	1	50%
Facilitate the use of the hand in other activities (computer keyboard, tablets…)	UL	0	0	0%
Facilitate the wearing of a resting orthosis	UL	0	0	0%
Facilitate access to the perineum (probes, sexuality, toilet, hygiene…)	LL	0	0	0%

Legend: LL = lower limb; UL = upper limb.

## Data Availability

Data are available from the corresponding author upon reasonable request.

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
