# Peer review of "Goal-Setting in Multiple Sclerosis-Related Spasticity Treated with Botulinum Toxin: The GASEPTOX Study"

_toxins, 2022, doi:10.3390/toxins14090582_

Round 1

Reviewer 1 Report

Comments to the Authors

Abstract:

Line 4 and 5: please review sentence to say that there is a "lack of evidence" rather than "weak evidence". Or explain that the evidence should be improved.

Introduction:

Line 32: space after "disease"

Line 34: please provide mortality rate of the MS disease

Line 59: please clarify if it was a lack of evidence or a lack of precise and conclusive evaluation?

Line 75: please provide the authors names before (22)

Line 82-83: Please clarify the sentence since "could implement" is not clear.

Methods:

Line 93-94: please rephrase since the BTI sessions cannot be eligible.

Line 96: space after "of"

Line 100: please explain if the goal definition method presented in this paper is innovative?

Line 118: Main data "consisted of "

Line 128: "pre/post" spasticity

Line 133: Could you indicate if the patients signed an informed consent form?

Results: 

Line 144: please give details on the antispastic treatments taken by the patients

Table 1: Why only 2 centers are presented since 3 hospitals were involved in the study (line 91).

Could you explain why female patients are more represented?

Line 152-155: please clarify why goal setting did not aim at the upper limb too?

Table 2: Please give details on the percentage calculations. Was it a percentage of patients showing improvement?

Table 3: please provide the definition of goal 1, goal 2 and goal 3 at the bottom of the table.

Table 4: please review alignement of the EDSS percentages

Table 5: idem: please review alignement of the % for center 1 versus center 2 and for the EDSS (6-7.5)

Table 6: prevent or limit retraction is listed twice.

Please explain the difference between selected times and achieved times.

Figure 1: Is it possible to show control data of non-treated patients that would be assessed in parallel?

Discussion:

Line 215: Please explain in more detail how the objectives differ from post-stroke patients.

line 239-243: please provide some percentages as example of good improvements.

Line 258-259: please indicate if it may be possible to reduce the number of goals because a SMART approach is probably easier with less criteria.

Line 263: "prevented"

Line 265: "designed"

Line 268-269: conclusions could be drawn even if the goals were not completely standardized

Author Response

REVIEWER 1

Comments to the Authors

Abstract:

Line 4 and 5: please review sentence to say that there is a "lack of evidence" rather than "weak evidence". Or explain that the evidence should be improved.

Introduction:

Line 32: space after "disease"

Line 34: please provide mortality rate of the MS disease

  • The purpose of this study is to explore functional objectives for BTI. The overall goal of the GASEPTOX project would be to demonstrate that in MS, treating focal spasticity could help function (active or passive) and in fine, patients’ quality of life. We surely won’t aim to reduce mortality.

Line 59: please clarify if it was a lack of evidence or a lack of precise and conclusive evaluation?

  • Rather a lack of evidence or evidence that has not been properly demonstrated. Especially for post stroke spasticity. This has been clarified.

Line 75: please provide the authors names before (22)

  • Thanks for pointing out this typo. It has been added in the revised version

Line 82-83: Please clarify the sentence since "could implement" is not clear.

  • This has been clarified.

Methods:

Line 93-94: please rephrase since the BTI sessions cannot be eligible.

  • This has been rephrased

Line 96: space after "of"

  • Modified accordingly

Line 100: please explain if the goal definition method presented in this paper is innovative?

  • The goal definition has been made according to usual standards and known literature, especially for post stroke spasticiy. Nothing “innovative”. What is innovative here is the fact that this was done, collected and analysed for patients with MS.

Line 118: Main data "consisted of "

  • Modified accordingly

Line 128: "pre/post" spasticity

  • Modified accordingly

Line 133: Could you indicate if the patients signed an informed consent form?

  • According to French legislation, for this kind of clinical research (MR004) a signed consent form is not needed, but information has to be written and delivered both orally and on paper. Furthermore, patients are informed of their right to oppose their consent to the retrospective use of their data.

Results: 

Line 144: please give details on the antispastic treatments taken by the patients

  • 45 had baclofen. 2 had an association of dantrolen and baclofen. This has been added.

Table 1: Why only 2 centers are presented since 3 hospitals were involved in the study (line 91).

Could you explain why female patients are more represented?

  • This actually confirms that we have a representative population of MS.

Line 152-155: please clarify why goal setting did not aim at the upper limb too?

  • GS also aimed at the UL. However it was way less frequent than for LL. This is discussed in the discussion section.

Table 2: Please give details on the percentage calculations. Was it a percentage of patients showing improvement?

  • % are reported to the global number of goals which explains on table 2 the “low” percentages.

Table 3: please provide the definition of goal 1, goal 2 and goal 3 at the bottom of the table.

  • This has been specified

Table 4: please review alignement of the EDSS percentages

  • Modified accordingly

Table 5: idem: please review alignement of the % for center 1 versus center 2 and for the EDSS (6-7.5)

  • This has been modified as possible. Usually this will be set by copy editing if the paper is accepted for publication.

Table 6: prevent or limit retraction is listed twice.

  • One for each limb (lower and upper)

Please explain the difference between selected times and achieved times.

  • This refers to: how many times a specified goal was selected, and how many times the same goal was considered achieved.

Figure 1: Is it possible to show control data of non-treated patients that would be assessed in parallel?

  • This would indeed be very interesting. However this was a retrospective study and not a controlled trial. No control group analysis could be performed.

Discussion:

Line 215: Please explain in more detail how the objectives differ from post-stroke patients.

  • In post stroke patients, the most studied population in this field, studied focusd on UL. When combining UL and LL, patients tended to select preferentially passive function goals. In our population, LL goals prevailed on UL goals and concerned mostly active function (walking+++). This has been clarified and detailed.

line 239-243: please provide some percentages as example of good improvements.

  • We detailed this paragraph adding some highlight examples

Line 258-259: please indicate if it may be possible to reduce the number of goals because a SMART approach is probably easier with less criteria.

  • We totally agree on that. This has been mentioned accordingly

Line 263: "prevented"

  • Modified accordingly

Line 265: "designed"

  • Modified accordingly

Line 268-269: conclusions could be drawn even if the goals were not completely standardized

  • We meant that we could not draw conclusions regarding BTI functional efficacy, not only due to the lack of standardization, but mostly because the study was not designed as a clinical trial with this objective in mind.

Reviewer 2 Report

This retrospective study on the patient's goals after BoNT-A injection deal with an interesting field.

My suggestions and queries as in queue:

- appears strange that in a retrospective study the ethic approval date is before to the target year for the study.

- The statement reported in the abstract and in the text "there is weak evidence of a functional improvement following BTI" deserve a little more explanation in the text.

- The goals appears well explain but almost no data are reported on the BoNT-A injection although through witch the patient can get the achievement of the goal.

- The Authors taking into account also goals referring to the upper limb but no functional data are reported for this one but only EDSS is listed in table 1 which is known to be based on the lower limb function.

Author Response

REVIEWER 2

This retrospective study on the patient's goals after BoNT-A injection deal with an interesting field.

My suggestions and queries as in queue:

- appears strange that in a retrospective study the ethic approval date is before to the target year for the study.

> In France, for this “méthodologie de reference”, an approval number is provided for a department or an institution, and assess that it complies to methodological standards (here for retrospective study). As this methodology do not imply a therapeutic intervention in a clinical trial, we only need to prove that we comply to this MR004 agreement and declare to the CNIL that we will use data accordingly.

Hence, an approval number can be provided and retrospective studies performed after.

- The statement reported in the abstract and in the text "there is weak evidence of a functional improvement following BTI" deserve a little more explanation in the text.

> This has been pointed out by the reviewer #1, and clarified accordingly.

- The goals appears well explain but almost no data are reported on the BoNT-A injection although through witch the patient can get the achievement of the goal.

> This is because we do not focus on BTI per se, but rather on what goals are set by patients and their physicians.

- The Authors taking into account also goals referring to the upper limb but no functional data are reported for this one but only EDSS is listed in table 1 which is known to be based on the lower limb function.

> This is because we did not aim at describing our population, but to know what are the most frequently selected goals when patients are followed in a referral spasticity clinic. This is interesting to note that even if UL function can be impaired early in the disease course, as long as locomotor abilities are (partly) preserved, LL and walking goals prevail on UL and passive function goals such as hygiene.

“We hereby demonstrated that lower limb function may be the main concern for patients with MS and their physicians, until complete loss of locomotor abilities, even if upper limb function could already be impaired” lines 223-225 in the revised version

Reviewer 3 Report

In this manuscript, the authors described a results of one-year multi-centre retrospective observational study assessing goal-setting and achievement during BTI session in spastic patients with multiple sclerosis.

 This manuscript is interesting; unfortunately, the manuscript needs substantial improvements and corrections before publishing may be possible.

General points:

Please check whole manuscript for spells.

Please add a list of abbreviations before References section to your manuscript.

Please check all dots at the end of each sentence. 

Please say in the whole manuscript: Table 1, 2, 3, 4, 5, 6.

Special points

Importantly, this manuscript should be substantially improved, i. e., by substantial references in the field.

Key contribution: please correct this words combination.

Introduction

Lines 28-30: please add multiple references at the end of this sentence.

Lines 30-31: please add multiple references at the end of this sentence.

Lines 31-34: please add more references at the end of this sentence.

Lines 34-35: please add multiple references at the end of this sentence.

Lines 35-38: please add more references at the end of this sentence.

Lines 38-46: please add multiple references at the end of each of these sentences.

Lines 47-52: please add multiple references at the end of each of these sentences.

Lines 53-61: please add multiple references at the end of each of these sentences and please describe very exactly all these studies.

Lines 63-65: please add multiple references at the end of each of these sentences.

Lines 63-71: please describe very exactly all these studies.

Methods

Lines 99-108: please describe this section more exactly.

Lines 110-112: please describe all formulations very exactly.

Lines 118-124: please add the appropriate references for all scales.

Discussion

Important: please discuss all your results very exactly and step by step.

Lines 213-221: please describe all these studies very exactly.

Conclusions

Please add also the Future perspectives section.

Author Response

REVIEWER 3

In this manuscript, the authors described a results of one-year multi-centre retrospective observational study assessing goal-setting and achievement during BTI session in spastic patients with multiple sclerosis.

 This manuscript is interesting; unfortunately, the manuscript needs substantial improvements and corrections before publishing may be possible.

General points:

Please check whole manuscript for spells.

  • Spells and typos have been checked

Please add a list of abbreviations before References section to your manuscript.

  • This has been added accordingly

Please check all dots at the end of each sentence. 

  • This has been checked

Please say in the whole manuscript: Table 1, 2, 3, 4, 5, 6.

  • Tables are referred within the main text as “table” with corresponding number

Special points

Importantly, this manuscript should be substantially improved, i. e., by substantial references in the field.

Key contribution: please correct this words combination.

  • It has been rephrased.

Introduction

Lines 28-30: please add multiple references at the end of this sentence.

Lines 30-31: please add multiple references at the end of this sentence.

Lines 31-34: please add more references at the end of this sentence.

Lines 34-35: please add multiple references at the end of this sentence.

Lines 35-38: please add more references at the end of this sentence.

Lines 38-46: please add multiple references at the end of each of these sentences.

Lines 47-52: please add multiple references at the end of each of these sentences.

Lines 53-61: please add multiple references at the end of each of these sentences and please describe very exactly all these studies.

Lines 63-65: please add multiple references at the end of each of these sentences.

Lines 63-71: please describe very exactly all these studies.

  • We understand that the list of bibliographical references should precise and relevant. Please, take into account that this is not a PhD report, nor a systematic review of literature. We therefore selected the references that appeared to us as the more relevant and highlighting our purpose.
  • If reviewer #3 feels that a particular reference in the studied field is lacking, we will be glad to add it (them) as suggested.
  • Finally, please note that when we cite Lance or Pandyan, the princeps paper is sufficient.

Methods

Lines 99-108: please describe this section more exactly.

  • Substantial modifications were already made according to both other reviewers’ comments.

Lines 110-112: please describe all formulations very exactly.

  • We clarified this with : abobotulinum toxin, incobotulinum toxin and onabotulinum toxin, without stating commercial names.

Lines 118-124: please add the appropriate references for all scales.

  • EDSS and GAS were right. We added the ref for the modified ashworth scale.

Discussion

Important: please discuss all your results very exactly and step by step.

Lines 213-221: please describe all these studies very exactly.

  • Again, rather than writing a systematic review of literature, we used referred studies to highlight our discussion. Please take into account that the actual revised manuscript is already 16 pages-long and propose 44 bibliographical references.

Conclusions

Please add also the Future perspectives section.

  • Although already mentioned in the discussion, a “future perspectives” section has been added accordingly.

Round 2

Reviewer 3 Report

Dear authors, thank you for your corrections. 

Unfortunately, this manuscript needs once more corrections before publishing may be possible.

General points:

 Once again, please check all dots at the end of each sentence. 

Once again, please say in the whole manuscript: Table (with capital letters).

Special points:

Once again, this manuscript should be substantially improved by substantial references in the field.

Key contribution: please say: Key contribution and also in bold.

Introduction

Lines 75-81: please add multiple references at the end of each these sentences.

Line 83: please add references at the end of this sentence.

Lines 92-150: please add references at the end of this sentence.

Lines 151-159: please add more references at the end of each these sentences.

Lines 160-162: please add references at the end of this sentence.

Lines 167-169: please add references at the end of this sentence.

Author Response

We thank the reviewer 3 for his/her very thorough reading and reviewing of our article. We tried to respond to the best of our possibilities. Please find below a point by point response. We understand that references may be lacking to rev. 3 point of view. However we chose what we believed to be the most relevant references. If one in particular is lacking please tell us.

Dr Levy and coll.

General points:

 Once again, please check all dots at the end of each sentence. 

  • Sentences were checked and dots put at the end when forgotten.

Once again, please say in the whole manuscript: Table (with capital letters).

  • Table has been written with a capital T as required.

Special points:

Once again, this manuscript should be substantially improved by substantial references in the field.

Key contribution: please say: Key contribution and also in bold.

  • Contribution was put in bold

Introduction

Lines 75-81: please add multiple references at the end of each these sentences.

  • We understand that as the use of the GAS methods in rehabilitation science is increasing and generalized, lots of references could be added. Hence, almost half of F Khan, Ashford & Turner-Stokes or Krasny-Pacini ‘s bibliography could be cited. We believe that we have chosen the most relevant. 7 references were cited to highlight and justify this paragraph [23-29]. We accept that we may have missed one particular that the reviewer wish to see cited. If so, please provide us with it and we will be glad to add it. Meanwhile we reorganized the references already cited.

Line 83: please add references at the end of this sentence.

  • We justified our hypothesis as follows:

Different from stroke, MS is a progressive disease with multiple functional consequences of related spasticity, and affecting mostly young adults [9]. We first hypothesized that objectives for the treatment of focal spasticity with intramuscular BTI, may differ for patients with MS compared to post-stroke patients.

Lines 92-150: please add references at the end of this sentence.

Lines 151-159: please add more references at the end of each these sentences.

Lines 160-162: please add references at the end of this sentence.

Lines 167-169: please add references at the end of this sentence.

  • For the latter, we really do not understand the comment here. The reviewer refers to the results section.